# Learning-based Preference Prediction for Constrained Multi-Criteria Path-Planning

**Kevin Osanlou**[1,3], **Christophe Guettier**[1], **Andrei Bursuc**[2], **Tristan Cazenave**[3], **Eric Jacopin**[4],

[1] Safran Electronics & Defense

[2] valeo.ai

[3] LAMSADE, Université Paris-Dauphine

[4] CREC, Ecoles de Saint-Cyr Coëtquidan

kevin.osanlou@safrangroup.com christophe.guettier@safrangroup.com andrei.bursuc@valeo.com
tristan.cazenave@lamsade.dauphine.fr eric.jacopin@st-cyr.terre-net.defense.gouv.fr

## Abstract

Learning-based methods are increasingly popular for search algorithms in single-criterion optimization problems. In contrast, for multiple-criteria optimization there are significantly fewer approaches despite the existence of numerous applications. Constrained path-planning for Autonomous Ground Vehicles (AGV) is one such application, where an AGV is typically deployed in disaster relief or search and rescue applications in off-road environments. The agent can be faced with the following dilemma : optimize a source-destination path according to a known criterion and an uncertain criterion under operational constraints. The known criterion is associated to the cost of the path, representing the distance. The uncertain criterion represents the feasibility of driving through the path without requiring human intervention. It depends on various external parameters such as the physics of the vehicle, the state of the explored terrains or weather conditions. In this work, we leverage knowledge acquired through offline simulations by training a neural network model to predict the uncertain criterion. We integrate this model inside a path-planner which can solve problems online. Finally, we conduct experiments on realistic AGV scenarios which illustrate that the proposed framework requires human intervention less frequently, trading for a limited increase in the path distance.

## 1 Introduction

Operations carried out by an autonomous ground vehicle (AGV) are constrained by terrain structure, observation abilities, embedded resources. For instance, in disaster relief operations or area surveillance, maneuvers must consider terrain knowledge. In most cases, the AGV ability to maneuver in its environment has direct impact on operational efficiency. Several perception capabilities (online mapping, geolocation, optronics, LIDAR) enable it to update its environment awareness online. Different mission planning layers can then provide continued navigation plans which are used for controlling the robotic platform, enabling it to fulfill a set mission. The AGV automatically manages its trajectory and follows navigation waypoints using control and time sequence algorithms.

Such mission planners could integrate A* algorithms (Hart, Nilsson, and Raphael 1968) as a best-first search

approach in the space of available paths. For a complete overview of static algorithms (*e.g.*, A*), replanning algorithms (*e.g.*, D*), anytime algorithms (*e.g.*, ARA*), and anytime replanning algorithms (*e.g.*, AD*), we refer the reader to (Ferguson, Likhachev, and Stentz 2005). Algorithms stemming from A* can handle some heuristic metrics but can become complex to develop when dealing with several constraints simultaneously like mandatory waypoints and distance metrics, or several optimization criteria. Our approach is based on Constraint Programming (CP). CP provides a powerful baseline to model and solve combinatorial and / or constraint satisfaction problems (CSP). It has been introduced in the late 70s (Laurière 1978) and has been developed until now (Hentenryck, Saraswat, and Deville 1998; Ajili and Wallace 2004; Carlsson 2015), with several real-world autonomous system applications, in space (Bornschlegl, Guettier, and Poncet 2000; Simonin et al. 2015), aeronautics (Guettier and Lucas 2016) and defense (Goldman et al. 2002).

Nevertheless, difficult weather and off-road conditions can make it impossible for an AGV to proceed through some paths autonomously. The intervention of a human, either onboard or with a remote control system, would be required in such situations. However, the purpose of an AGV is to reduce the crew workload in the first place. Coming up with a navigation plan which requires minimum human intervention and remains acceptable in terms of a metric such as the total distance is therefore critical. This work focuses on this particular issue, especially in a context where the decision criterion for human intervention is uncertain and needs to be learned offline. We refer to this uncertain criterion as *autonomous feasibility*. In our approach, we learn this criterion to assess where human intervention would be needed.

To this purpose, we use a simulated AGV environment that includes a decision function for this criterion as part of a higher-level environment representation. Computing the autonomous feasibility by running this function for online mission planning would require simulating the entire environment. This is impractical for real-time applications. Instead, we train a neural network to approximate this criterion offline. When planning the mission online, we use the neural network to make predictions for the autonomous feasibility. We define an optimization strategy for a CP-based planner and use it to compute navigation plans. We conduct

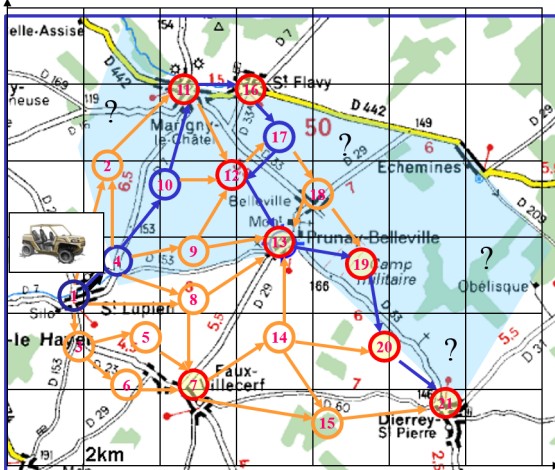

Figure 1: Search and rescue mission. Potential paths for an AGV throughout a flooded area between the Seine and the Vanne rivers in the area of Troyes, France. The blue area represents the expected flooded area. Orange, red and blue circles represent different positions the AGV can proceed to. Red circles are possible waypoints while blue circles are mandatory ones. Orange edges represent trafficability and blue ones a potential optimal solution in terms of distance cost.

experiments on realistic scenarios and show that the proposed framework reduces need for human interactions, trading with an acceptable increase of the total path distance.

## 2  Context and Problem Formalization

AGV operations require online path-planning, done accordingly with the demands of an operator who defines the objectives of a mission. For instance, in disaster relief, the vehicle has to perform some reconnaissance tasks in specific areas. Figure 1 shows a flooded area and possible paths to assess disaster damages. Possible paths are defined using a graph representation, where edges and nodes represent respectively ground mobility and accessible waypoints. Graphs are defined during mission preparation by terrain analysis and situation assessment. The AGV system responds to an operator demand by finding a route from a starting point to a destination, and by maneuvering through mandatory waypoints. During the mission, some paths may not be trafficable or new flooded areas to investigate may be identified, requiring immediate replanning. Additionally, some paths may prove difficult for the AGV given weather and road conditions and require the crew to take control of the vehicle to proceed forward. Aside from the terrain structure itself, these issues are mainly caused by changes in weather conditions:

- Rainfall: heavy rain may cause new muddy areas to occur, affecting cross-over duration between two waypoints or increasing the risk of losing platform control.
- Fog: heavy fog may cause a performance drop of the LI-DAR, making it harder for the AGV to keep track of the surrounding environment.

- Wind: heavy winds may require a high level of corrections in the followed trajectories.

In Figure 1, the AGV starts from its initial position (position 1). Areas in blue are flooded and the disaster perimeter must be evaluated by the vehicle. All nodes circled in red have to be visited, *e.g.*, refugees and casualties are likely to be found there. A typical damage assessment would require up to 10 mandatory nodes to visit. The first criterion is the global traverse distance that meets all visit objectives, and must be minimized. The second criterion is the autonomous feasibility, which needs to be maximized so as to reduce the likelihood of requiring human intervention.

Let $\mathcal{G} = (V, E)$ be a connected weighted graph. Each edge has a weight representing a distance metric. In addition to this weight, a binary feature represents the autonomous feasibility of the edge, which is not known a priori and depends on both weather conditions and terrain structure. While terrain structure mostly depends on the graph $\mathcal{G}$, the weather affecting the AGV is defined by three variables:

- $x_1 \in \{0, 1, ..., 10\}$ is the intensity of rainfall,
- $x_2 \in \{0, 1, ..., 10\}$ is the thickness of the fog,
- $x_3 \in \{0, 1, ..., 10\}$ is the strength of the wind

A typical instance $I$ of the path-planning problem we consider is defined as follows:

$$I = (s, d, M)$$

where:

- $s \in V$ is the start node in graph $\mathcal{G}$,
- $d \in V$ is the destination node in graph $\mathcal{G}$,
- $M \subset V$ is a set of distinct mandatory nodes that need to be visited at least once, regardless of the order of visit.

In order to solve instance $I$, the AGV has to find a path from node $s$ to node $d$ that passes by each node in $M$ at least once. There is no limit to how many times a node can be visited in a path. The solution path should compromise between minimizing the total path distance and maximizing the autonomous feasibility.

## 3  Environment Simulation

The environment in which the AGV evolves is modelled in a 3D map. Vertices defined in $\mathcal{G}$ represent positions in the environment, while edges are represented by a set of continuous sub-positions linking vertices. Figure 2 shows a typical environment simulated by the 4d Virtualiz software in which the AGV proceeds. The simulation is realistic as it takes into account not only vehicle physics, but more importantly the vehicle's sensors, as well as core programs. Among such programs lie the environment-building functions, which enable the vehicle to build a state of its surrounding environment from its sensors. Key functions such as obstacle avoidance or waypoint-follow functions are also implemented in the simulator mimicking real life situations with high fidelity. We leverage the simulated environment to design and test out our autonomous system in scenarios similar to real life conditions.

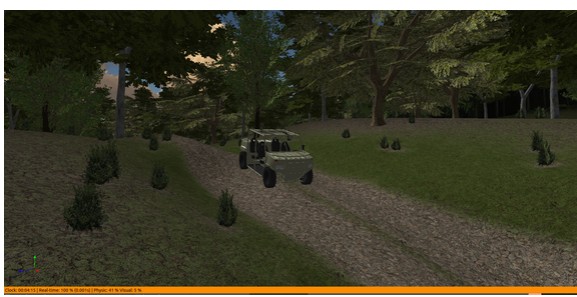

Figure 2: An AGV evolving in an environment created by the 4d Virtualiz simulator.

When a graph is defined, the simulator links it to the 3D map and several built-in functionalities become available. We can thus generate custom missions for the AGV by defining a start position, an end position, and a list of mandatory waypoints. Additional environment parameters are available in such simulators and we consider a few representative ones in this work. In particular, we experiment with the rain variable $x_1$, the fog variable $x_2$ and the wind variable $x_3$. When sending the AGV on a defined mission $I = (s, d, M)$ taking a path $P$, the simulating system will make the vehicle drive on a set of edges $e \in P$. Edges resulting in autonomous failure (vehicle getting stuck or taking longer than a set time-out threshold) $Q \subset P$ correspond to difficult sections of the path, which would require manual control of the vehicle. The criteria for autonomous feasibility depends on both current weather conditions and terrain structure and topology, as well as the vehicle's autonomous capabilities.

In order to learn to approximate the autonomous feasibility criterion, we use this simulated environment to create training data. To this end, we generate different weather conditions by changing the variables $x_1, x_2, x_3$ and send the vehicle on random missions. For a given set of values $x_1, x_2, x_3$ and a path $P$ that the vehicle has to follow, we retrieve the list of edges $Q \subset P$ which the simulator judges difficult for autonomous maneuvers. For each edge $e_i \in Q$, we store in a dataset $D$ a feature vector $\mathbf{x}_i = [x_1, x_2, x_3, \max(\text{slope}_i), \text{dist}_i]$ and $y_i = 0$. Variable $\max(\text{slope}_i)$ is the maximum slope of edge $e_i$, $\text{dist}_i$ the distance of edge $e_i$ and $y_i$ is the preference label associated with $\mathbf{x}_i$. With $y_i = 0$, the edge should be avoided if possible when planning under these weather conditions. Similarly for each edge $e_i \in P \backslash Q$, we store the value $\mathbf{x}_i$ and $y_i = 1$. These edges should be preferred when planning under these conditions. Regarding the structure of the terrain, we are unable to retrieve more information for an edge than only its distance and maximum slope. While this lack of information results in a limitation for learning performance, our experiments show that the performance of our framework remains satisfying.

## 4 Neural Network Training

In this section, we first provide a brief introduction to neural networks and then describe the manner we leverage them for our problem. We train the neural network for identifying the difficult areas on the map using terrain and weather information and supervision from the decisions of the simulator concerning the respective sections on the map.

Neural networks (NNs) allow computing and learning of multiple levels of abstraction of data through models with millions of trainable parameters. It is known that a sufficiently large neural network can approximate any continuous function (Funahashi 1989). Although the cost of training such a network can be prohibitive, modern training practices for multi-layer networks usually allow reasonable approximations for a large variety of problems. With this in mind, we attempt to train a neural network to approximate the decisions of the 3D simulator over edges of the graph map.

### 4.1 Neural Networks

Recently NNs, in particular deep neural networks, made a comeback in the research spotlight after achieving major breakthroughs in various areas of computer vision (Krizhevsky, Sutskever, and Hinton 2012), (Simonyan and Zisserman 2014), (He et al. 2016), (Ren et al. 2015), (Redmon et al. 2016), (Long, Shelhamer, and Darrell 2015), neural machine translation (Sutskever, Vinyals, and Le 2014), computer games (Silver et al. 2016) and many more fields. While the fundamental principles of training neural networks are known since many years, the recent improvements are due to a mix of availability of large image datasets, advances in GPU-based computation and increased shared community effort.

In spite of the complex structure of a NN, the main mechanism is rather straightforward. A *feedforward neural network*, or *multi-layer perceptron (MLP)*, with $L$ layers describes a function $f(\mathbf{x}; \boldsymbol{\theta}) : \mathbb{R}^{d_\mathbf{x}} \mapsto \mathbb{R}^{d_y}$ that maps an input vector $\mathbf{x} \in \mathbb{R}^{d_\mathbf{x}}$ to an output vector or scalar value $y \in \mathbb{R}^{d_y}$. Vector $\mathbf{x}$ is the input data that we need to analyze (*e.g.*, an image, a graph, a feature vector, etc.), while $\mathbf{y}$ is the expected decision from the NN (*e.g.*, a class index, a scalar, a heatmap, etc.). The function $f$ performs $L$ successive operations over the input $\mathbf{x}$:

$$h^{(l)} = f^{(l)}(h^{(l-1)}; \theta^{(l)}) = \sigma\left(\theta^{(l)} h^{(l-1)} + b^{(l)}\right) \quad (1)$$

where $h^{(l)}$ is the hidden state of the network and $f^{(l)}$ is the mapping function performed at layer $l$ and parameterized by trainable parameters $\theta^{(l)}$ and bias $b^{(l)}$, and piece-wise activation function $\sigma(\cdot)$; $h^{(0)} = \mathbf{x}$. We denote by $\boldsymbol{\theta} = \{\theta^{(1)}, \dots, \theta^{(L)}\}$ the entire set of parameters of the network. Intermediate layers are actually a combination of linear classifiers followed by a piece-wise non-linearity from the activation function. Layers with this form are termed *fully-connected layers*.

NNs are typically trained with labeled training data, *i.e.* a set of input-output pairs $(\mathbf{x}_i, y_i)$, $i = 1, \dots, N$, where $N$ is the size of the training set. During training we aim to minimize the training loss:

$$\mathcal{L}(\boldsymbol{\theta}) = \frac{1}{N} \sum_{i=1}^{N} \ell(\hat{y}_i, y_i), \quad (2)$$

where $\hat{y}_i = f(\mathbf{x}_i; \boldsymbol{\theta})$ is the estimation of $y_i$ by the NN and $\ell : \mathbb{R}^{d_L} \times \mathbb{R}^{d_L} \mapsto \mathbb{R}$ is the loss function. The loss $\ell$ measures the distance between the true label $y_i$ and the estimated one $\hat{y}_i$. Through *backpropagation* (Rumelhart et al. 1988), the information from the loss is transmitted to all $\boldsymbol{\theta}$ and gradients of each $\theta_l$ are computed w.r.t. the loss $\ell$. The optimal values of the parameters $\boldsymbol{\theta}$ are then found via stochastic gradient descent (SGD) which updates $\boldsymbol{\theta}$ iteratively towards the minimization of $\mathcal{L}$. The input data is randomly grouped into mini-batches and parameters are updated after each pass. The entire dataset is passed through the network multiple times and the parameters are updated after each pass until reaching a satisfactory optimum.

## 4.2 Training Setup

We define a neural network $f$ that takes as input a vector $\mathbf{x}_i = [x_1, x_2, x_3, \max(\text{slope}_i), \text{dist}_i]$ and outputs the probability $\hat{y}$ of the edge $e_i$ being a preferred edge for autonomous navigation or not. The network $f$ consists of 4 fully-connected layers interleaved with *ReLU* non-linearities and with a *sigmoid* activation at the end. The output of the sigmoid $\in [0, 1]$ is rounded to the closest integer 0 or 1 when classifying an edge under given weather and terrain conditions.

The simulator serves as *teacher* to the NN, which learns here to mimic the simulator's decisions based on the path configuration and weather. Following the creation of dataset $D$ in section (§ 3) we use the pairs of edges and labels $(\mathbf{x}_i, y_i)$ to train the neural network to correctly predict the label node $\hat{y}_i$. For supervision we use the binary cross-entropy loss, typically used for binary classification tasks:

$$\ell(\hat{y}_i, y_i) = -[y_i \log \hat{y}_i + (1 - y_i) \log(1 - \hat{y}_i)] \quad (3)$$

We train the NN using SGD with momentum. In order to prevent overfitting, we do *early stopping*, *i.e.*, we halt the training once the average loss on the validation set stops decreasing and starts increasing. In our experiments, the neural network $f$ achieves an accuracy of 79% on the validation set. This is due to the lack of features which come into play in deciding the autonomous feasibility for an edge. In section (§6), this performance is tested and evaluated on new situations. In comparison, we also ran a logistic regression on the same training set, which achieved a validation accuracy of 71%. We believe some feature engineering may be necessary to provide slightly more relevant features for the logistic regression.

# 5 Constrained Multi-Criteria Optimization for Navigation and Maneuver Planning

We aim to compute an optimized navigation plan in cross-country areas. In our approach, the navigation plan is represented as a path sequence of waypoints in a predefined graph. A "good" plan must minimize distance and maximize autonomous feasibility while satisfying mandatory waypoints.

Path planning is achieved using Constraint Programming (CP), here with a model-based constraint solving approach (Guettier and Lucas 2016) and cost objective functions (equations 7 and 8). The problem is formulated in CP as a Constraint Optimization Problem (COP). Distance and autonomous feasibility are considered as primary and secondary cost objectives, respectively. We propose a multi-criteria optimization algorithm, based on global search, and adapted from branch and bound (B&B) techniques (Narendra and Fukunaga 1977).

Both COP formulation and search techniques are implemented with the CLP(FD) domain of SICStus Prolog library (Carlsson 2015). It uses the state-of-the-art in discrete constrained optimization techniques and Arc Consistency-5 (AC-5) (Deville and Van Hentenryck 1991; Van Hentenryck, Deville, and Teng 1992) for constraint propagation, implemented as CLP(FD) predicates.

The search technique is hybridized with a probing method (Guettier and Lucas 2016), allowing automatic structuring of the global search tree. In this paper, probing focuses on learned and predicted autonomous feasibility in order to define an upper bound to the secondary metric. Probing takes as input predicted autonomous feasibility, builds up a heuristic sub-optimal path based on it as a preference, and lastly initializes the secondary cost criterion. The resulting algorithm is a Probe-based Constraint Multi-Criteria Optimizer denoted as PCMCO.

## 5.1 Planning Model with Flow Constraints for Multi-Criteria Optimization

The PCMCO elaborates a classical flow formulation with integrals, widely used in operation research (Gondran and Minoux 1995). For a given path-planning problem $I = (s, d, M)$, the set of possible paths is modelled as a graph $\mathcal{G} = (V, E)$, where $V$ is the set of vertices and $E$ the set of elementary paths between vertices. A set of flow variables $\varphi_e \in \{0, 1\}$, where $e \in E$, models a possible path from $start \in V$ to $end \in V$. A flow variable for an edge $e = (v, v')$ is denoted as $\varphi_{vv'}$. An edge $e$ belongs to the navigation plan if and only if $\varphi_e = 1$. The resulting navigation plan is represented as $\Phi = \{e | e \in E, \varphi_e = 1\}$. From an initial position to a requested final one, path consistency is enforced by flow conservation equations, where $\omega^+(v) \subset E$ and $\omega^-(v) \subset E$ represent respectively outgoing and incoming edges from vertex $v \in V$. Since flow variables are $\{0, 1\}$, equation (4) ensures path connectivity and uniqueness while equation (5) imposes limit conditions for starting the path at $s$ and ending it at $d$:

$$\sum_{e \, \in \, \omega^+(v)} \varphi_e = \sum_{e \, \in \, \omega^-(v)} \varphi_e \leq N \quad (4)$$

$$\sum_{e \, \in \, \omega^+(s)} \varphi_e = 1, \quad \sum_{e \, \in \, \omega^-(d)} \varphi_e = 1, \quad (5)$$

These constraints provide a linear chain alternating pass-by waypoint and navigation along the graph edges. Constant $N$ indicates the maximum number of times the vehicle can pass by a waypoint. With this formulation, the plan may contain cycles over several waypoints. Mandatory waypoints are imposed using constraint (6). Path length is given by the metric (7), and we will consider the path length as

the primary optimization $D_{end}$ criterion to minimize, where constants $d_{vv'}$ represent elementary path distance between vertices. They are provided off-line, at mission preparation time. Likewise, the secondary criterion $P_{end}$ has the same formulation and is based on autonomous feasibility (8). In turn, constants $p_{vv'}$ are edge preferences resulting from the predictions of the neural network $f$ on autonomous feasibility for each edge $e \in E$.

$$\forall v \in M \sum_{e \ \in \ \omega^+(v)} \varphi_e \geq 1 \qquad (6)$$

$$\forall v \in V, D_v = \sum_{v'v \ \in \ \omega^-(v)} \varphi_{v'v} d_{vv'} \qquad (7)$$

$$\forall v \in V, P_v = \sum_{v'v \ \in \ \omega^-(v)} \varphi_{v'v} p_{vv'} \qquad (8)$$

## 5.2 Global Search Algorithm

The global search technique underlying PCMCO guarantees completeness, as well as proof of completeness. It is based on classical algorithmic components:

- Variable filtering with correct values, using specific labeling predicates to instantiate problem domain variables. AC-5 being incomplete, value filtering guarantees search completeness.

- Tree search with standard backtracking when instantiating a variable fails.

- Branch and Bound (B&B) for both primary and secondary cost optimization, using minimize predicate.

Within the B&B algorithm, the primary cost $D_{end}$ drives the optimization loop. We extend the algorithm with preference optimization $P_{end}$ to converge towards a pareto optimal solution. At each iteration $k$, we impose that $P_{end}^{k+1} \leq P_{end}^k$ as a secondary optimization schema. This constraint is weaker than $D_{end}^{k+1} < D_{end}^k$, classically applied to the primary distance cost, which corresponds to the default operational semantic predicate *minimize* of the SICStus Prolog library. The $P_{end}^0$ is initialized by probing with an arbitrary heuristic solution obtained with the Dijsktra algorithm. In this manner, B&B will favor learned preferences.

Note that in general probing techniques (Sakkout and Wallace 2000), the order can be redefined within the search structure (Ruml 2001). Similarly, in our approach, the variable selection order provided by the probe can still be iteratively updated by the labeling strategy that makes use of other variable selection heuristics. Mainly, first fail variable selection is used in addition to the initial probing order.

These algorithmic designs have already been reported with different probing heuristics (Guettier and Lucas 2016), such as A* or meta-heuristics such as Ant Colony Optimization (Lucas et al. 2010),(Lucas, Guettier, and Siarry 2009). However, other multi-criteria optimization techniques could be used, for instance based on valued constraint satisfaction problems (VCSP) (Schiex et al. 1995) or soft constraints (Domshlak et al. 2003). In our design, the search is still complete, guaranteeing proof of completeness, but demonstrates efficient pruning.

## 6 Experiments

For a given problem, minimizing the total distance of a solution path while maximizing autonomous feasibility are contradictory objectives requiring a compromise. This section carries two purposes. The first is to verify that the neural network $f$ is capable of making consistent predictions to avoid difficult edges. The second is to evaluate the compromise made by the CP-based solver described previously.

We generate 200 random benchmark instances associated with a graph $\mathcal{G}$ (Guettier 2007) that is representative of real scenarios for AGV search & rescue operations. We consider three different types of weather conditions: *fine*, *moderate* and *difficult*. For each weather type, we randomly select 50 instances, and we compare the solutions given by two solvers. The first solver is the reference probe-based constrained optimizer (PCO), which does not explore any preference criterion and only optimizes the distance. The second solver is the upgraded version with multi-criteria optimization (denoted as PCMCO). It takes into account the preference predictions of the neural network $f$ for current weather conditions. For each edge $e_i \in E$, the preference prediction is obtained with a forward pass of the feature vector described in section (§4). We denote the resulting hybridization as NN + PCMCO.

Table 1: Experiments carried out on benchmark instances of graph $\mathcal{G}$. The first metric reported is the distance of the solution path, in meters. The second one is the number of human interventions required in the solution path. For both metrics, we compute the mean, median (med) and standard deviation (std) over all benchmark instances.

| Weather & Method: | Distance (m) | | | Interventions | | |
|---|---|---|---|---|---|---|
| | mean | med | std | mean | med | std |
| Fine weather | | | | | | |
| PCO | 4463 | 4246 | 840 | 1.6 | 2 | 1.1 |
| NN + PCMCO | 4912 | 5016 | 954 | 0.1 | 0 | 0.3 |
| Moderate weather | | | | | | |
| PCO | 4166 | 4102 | 675 | 2.3 | 2 | 1.2 |
| NN + PCMCO | 5431 | 5390 | 1003 | 0.4 | 0 | 0.6 |
| Difficult weather | | | | | | |
| PCO | 4207 | 4115 | 687 | 4.1 | 4 | 1.2 |
| NN + PCMCO | 5153 | 5256 | 881 | 2.5 | 2 | 1.4 |

We study the influence of the edge preferences given by the neural network $f$ on the solution path. For each instance, we compute the solution path given by each solver. The total distance is then computed by summing the distances of all edges in the solution path. The solution path is also simulated in the 3D simulation environment to count the number of required human interventions. The human intervention count used in this section is a criterion which is opposite to the autonomous feasibility criterion, and should therefore be minimized. Results are averaged per instance and reported in table 1.

For fine weather conditions, the use of the neural network

preferences enables NN + PCMCO to almost never require human assistance in exchange for a $10\%$ higher distance cost than PCO's. On the other hand, PCO requires more than 1 human intervention per instance on average. For moderate weather conditions, we see those gaps widening. A $30\%$ higher distance cost allows NN + PCMCO to require far less human interventions than PCO. Lastly, for difficult weather conditions, we observe that NN + PCMCO incurs a $22\%$ higher distance cost. While NN + PCMCO requires far less human interventions than PCO, it still requires more than 2 human interventions on average. This is explained by the fact that difficult weather conditions cause a majority of edges to be difficult for autonomous driving. The solution path has no choice but to include some of those edges. This also explains the lower distance cost increase than for moderate weather conditions.

Additionally, we run statistical tests to compare PCO and NN+PCMCO and summarize them in table 2. Firstly, a paired sample t-test is done, for each weather condition, to compare the mean path distance given by PCO and NN+PCMCO. The high t-values obtained, combined with very low p-values, indicate that the distance costs found by PCO and NN+PCMCO differ significantly and that it is very unlikely to be due to coincidence. Secondly, a $\tilde{\chi}^2$ test is performed on the intervention count criterion for each weather condition. The high p-values observed validate the hypothesis that NN+PCMCO acts independently of PCO in terms of autonomous feasibility.

Table 2: Statistical tests run on benchmark results. The paired sample t-test is run on the distance criterion, while the $\tilde{\chi}^2$ test is run on the intervention count criterion.

| Test Method | Paired t-test | | $\tilde{\chi}^2$ test | |
| --- | --- | --- | --- | --- |
| | t-value | p-value | $\tilde{\chi}^2$-value | p-value |
| Fine weather | 7.28 | $10^{-9}$ | 19.1 | 0.99 |
| Moderate weather | 8.18 | $10^{-9}$ | 19.4 | 0.89 |
| Difficult weather | 6.51 | $10^{-7}$ | 9.37 | 0.99 |

These results highlight the fact that the neural network $f$ makes consistent predictions, and that NN + PCMCO offers a good compromise between distance metric and autonomous feasibility.

## 7 Conclusion

We introduced a method for online constrained path-planning problems with two optimization criteria, based on learned preferences. The distance criterion needs to be minimized, while the autonomous feasibility criterion, which is uncertain, has to be maximized. Our approach proposes offline learning of a model for autonomous feasibility in simulation environments. We also introduced a CP-based algorithm which takes into account the model's prediction of autonomous feasibility and compromises between both criteria for online path-planning. Experiments suggest the proposed framework is capable of finding a good compromise

which offers a higher autonomous feasibility for an acceptable increase in distance cost. AGV crew could benefit from such an approach, mostly in situations where their workload needs to be reduced.

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
