# OpenReview forum: "Learning-based Preference Prediction for Constrained Multi-Criteria Path-Planning"
_icaps-conference.org/ICAPS/2019/Workshop/SPARK — SPARK 2019_

### Official Review · AnonReviewer1 · 2019-04-30
**Review1**

**Rating:** 4
**Confidence:** 3

**Review:**

The paper describes a hybrid learning and planning problem for controlling ground vehicles that must visit a set of  waypoints in a challenging environment with varying weather conditions.  The system adapts its plans to a human operator with the aim of creating the best feasible schedule for the vehicles  while minimizing the number of interventions required by the human operator.  The environment has weather conditions (rainfall, wind, and fog) that vary the difficulty of traversing the area.  A Feed-forward Neural Network is trained to predict terrain difficulty based on data gathered from a high-fidelity simulator.  Separately, a network flow model is used to evaluate possible solutions for visiting the required waypoints.  In a set of experiments, the results demonstrate that the hybrid NN+optimization approach reduces the number of interventions while increasing the overall path distance.

The paper was mostly easy to read and well motivated.  Overall, I think it would make a nice contribution to the SPARK workshop.  I'll close with some suggestions/thoughts for the authors to clarify in their next revision.

The neural network seems less clearly motivated, especially since it seemed that statistical method might perform just as well.  In part, this is due to my suspicion that the features predicting terrain difficulty would be correlated (a strong storm probably produces high wind, heavy rain, for example).  I encourage the authors to considered using a statistical model as a baseline for understanding how much contribution the NN adds.

The paper could probably make a stronger argument for its claims if Table 1 included additional statistics such as the standard deviation and median.  There are also enough problem instances to analyze the differences between the approaches using a paired sample t-test (for path distance) a Chi-squared test (for required interventions) and an ANOVA  for comparing the effect of weather.

---

### Official Review · AnonReviewer2 · 2019-05-02
**An interesting paper with good results; presentation could be better**

**Rating:** 3
**Confidence:** 2

**Review:**

The paper proposes a constrained multi-criteria path-planning algorithm with path distance and human intervention as two objective functions. A neural network is built to predict the feasibility of driving through particular edges without human intervention. Such feasibility is then encoded in the path planning algorithm. The application and results are interesting.

Here are some suggestions on improving presentations:
- I read "user preferences" Section 5, but i had no clue what it is. Later I figured out it is linked to "autonomous feasibility" discussed in the previous sections. Please make this link more explicitly.
- in Section 2, instance I = (s,d,M) is defined, but s, d are not used in the formulation in Section 5.
- In Eqn (7,8), v'v is not defined.
- The optimization objective function is mentioned in the text. Can you write it down mathematically for the purpose of completeness of the formulation?
- The experiment could be elaborated more.  As the outputs of NNs are used as parameters of the optimization model, the performance of NNs in terms of accuracy should be discussed in the results. It is not clear what values of vector x are used, how many edges etc, in the experiments.
- The authors state learning-based methods have significantly few approaches for multi-criteria optimization, compared to single criterion optimization problems. As a reader, I would like to see some discussions in the paper on what could have been different on handling multi or single objectives in your problem? Is it a straightforward extension (from single objective), or does it involve additional technical challenges?

---

### Decision · Program_Chairs · 2019-05-08
**Acceptance Decision**

**Decision:**

Accept

**Comment:**

Good for a workshop paper, relevant for SPARK.